

# Temperature regulation in the Balkan spadefoot (*Pelobates balcanicus* Karaman, 1928) at the beginning of nocturnal activity

Nikolay Natchev[1,2], Teodora Koynova[1], Krasimir Tachev[1], Dimitar Doichev[1], Pavlina Marinova[1], Valeriya Velkova[3] and Daniel Jablonski[4]

[1] Shumen University, Shumen, Bulgaria
[2] University of Vienna, Vienna, Austria
[3] Unaffiliated, Shumen, Bulgaria
[4] Comenius University in Bratislava, Bratislava, Slovak Republic

Corresponding authors
Nikolay Natchev, natchev@shu.bg
Teodora Koynova,
t.koynova@shu.bg

## ABSTRACT

On land, the amphibians interact with the environment in a complex way-even small changes in the physiological conditions may significantly impact the behaviour and *vice versa*. In ectothermic tetrapods, the transition from inactive to active phase may be related to important changes in their thermal status. We studied the thermal ecology of adult Balkan spadefoots (*Pelobates balcanicus* Karaman, 1928) in northeastern Bulgaria. These toads spend the daytime buried between 10 and 15 cm in sandy substrates, and emerge after sunset. On the substrate, their thermal energy exchange is defined by the absence of heat flow from the sun. Secondary heat sources, like stored heat and infrared radiation from the soil play an important role for the thermal balance of the active spadefoot toads. At the beginning of their daily activity, we measured substrate temperature (at a depth of 11–12 cm), toad's surface body temperature, and also provided thermal profiles of the animals and the substrate surface in their microhabitats. In animals which recently emerged from the substrate, the temperature was comparatively higher and was closer to that of the subsoil on the spot. After that, body temperature decreased rapidly and continued to change slowly, in correlation with air temperature. We detected a temperature gradient on the dorsal surface of the toads. On the basis of our measurements and additional data, we discuss the eventual role of air humidity and the effects of surface and skin water evaporation on the water balance and activity of the investigated toads.

## INTRODUCTION

According to *Brattstrom (1979)*, it is crucial to relate changes in the body temperature to the behavioural changes in the study of thermoregulation in amphibians – "watching for emergence, and then recording body temperature of emerging animals". The ideas concerning the thermoregulation capacity of amphibians have changed dramatically with the advances of physiological science. According to modern concepts, body temperature

crucially impacts the biochemical and physiological processes in ectothermic organisms (see *Hutchison & Dupré, 1992*). These animals invest very little in heat production, hence amphibians are considered to have a low potential for increase in body temperature even at extreme levels of muscle activity. However, many species have a capacity at least for behavioural thermoregulation (see *Brattstrom, 1970*; *Lillywhite, Licht & Chelgren, 1973*; *Feder, 1992*). Any kind of thermoregulation may be related to very high costs (water loss, energy expenditure, predator pressure, *etc.*). Actually, minor changes in the thermoregulatory behaviour may have a significant impact on the mobility and habitat selection of the ectothermic tetrapods (*Bartelt & Peterson, 2005*; *Kearney, Shineb & Porterc, 2009*).

Most amphibians tend to retain thermal equilibrium with the surroundings. Terrestrial species like frogs and salamanders are rather small in size and are respectively constrained in their capacity for heat storage. Theoretically, such species should have an increased potential to thermoregulate by water evaporation when ambient air temperature is high and may reach a thermal equilibrium rather rapidly. The energy absorbed by the body should equate that which the body dissipates (see *Spotila, O'Connor & Bakken, 1992*). Actually, in amphibians, metabolism and thermoregulation are controlled by external factors such as water collection (*Lillywhite, 1970*) and the environmental temperature (*Mokhatla, Measey & Smit, 2019*). Heat transfer between the body and the environment may occur in different ways concerning the microhabitat (*Kreith, 1973*). Underwater, the most important processes for heat exchange in amphibians are convection and conduction. Temperature differences rarely exceed 2 °C (*Erskine & Spotila, 1977*). From a thermodynamic perspective, on land, the amphibians interact with the habitat in a more complex mode. The body exchanges thermo-radiation with the ground and the surrounding air. It interacts with the floating air due to convection, conducts heat from/to the ground, but also heats from different types of sun rays (direct, diffuse, reflected, or scattered). The body may cool predominantly by evaporating water.

Water evaporation plays a major role in lowering and maintaining a preferable body temperature at the cost of significant water loss (see *Tracy, 1976*; *Spotila, O'Connor & Bakken, 1992*). In fact, water equilibrium may play an even more important role in the fitness of amphibians than thermoregulation (see *Bartelt & Peterson, 2005*; *Titon & Gomes, 2015*; *Roznik, Rodriguez-Barbosa & Johnson, 2018*).

In the present study, we collected data on thermoregulation in *Pelobates balcanicus* – a marked terrestrial and strictly thigmothermic anuran species. We studied the thermal status of adult spadefoot toads during the change in their activity mode. As anurans possess a low potential to thermoregulate (*Spotila, O'Connor & Bakken, 1992*), we focused on the thermal conditions during which the Balkan spadefoots transition between two behavioural phases: rest and locomotion/hunting. The investigated toads spend the day buried in the substrate without direct exposure to solar radiation and hence their body temperature is solely determined by the thermal conditions of the substrate. During the active phase, *P. balcanicus* is situated in a system, which usually includes: the night sky, high air humidity, temperate sandy ground and air of comparatively constant temperature. We suggest that the combined effect of all these components may impact the

thermal biology and the long-term activity patterns of *P. balcanicus* in a rather complex manner.

## MATERIALS AND METHODS

### Ecological remarks on the studied species

The species belongs to a phylogenetically very old toad clade – a sister clade to the modern group Neobatrachia. This clade includes the families Scaphiopodidae, Pelodytidae, Megophryidae, and Pelobatidae (*Pyron & Wiens, 2011*). The close relationship of Pelobatidae with Scaphiopodidae (morphologically very similar toads, living in deserts of Northern America) suggests an old evolutionary association to fossorial lifestyle with a preference for lower body temperatures.

The Balkan spadefoot inhabits the Balkan Peninsula from the sea level up to 920 m a. s. l. (*Džukić et al., 2008*). In Bulgaria, the species distribution follows the Danube River and the river banks areas of Maritsa and Struma rivers. There are numerous populations along the northern Bulgarian Black sea coasts. The adults are active normally from February (but see *Koynova, Marinova & Natchev, 2022*) until November and live in open lowland areas. It is a strictly nocturnal species that remains hidden in burrows (10–15 cm deep) or under stones during the day (*Stojanov, Tzankov & Naumov, 2011*). The structure of the ground is a crucial factor in the habitat preferences of most Pelobatidae species, as most of them inhabit exclusively sandy soils (see *Eggert, 2002*; *Scali & Gentilli, 2003*; *Yermokhin, Ivanov & Tabachishin, 2013*).

### Field surveys, documentation, and measurements

The investigated population near Durankulak (northeastern Bulgaria) was visited during five evenings within 33 days in the late Spring and the early Summer of 2020 (14.06; 25.06; 27.06; 28.06; 17.07). We consistently started our field investigations at sunset before the adult spadefoot toads appeared on the surface. We always followed the same track and worked in one pair. Pursuant to *Stojanov, Tzankov & Naumov (2011)*, the investigated locality is inhabited by two species of spadefoot toads – *P. fuscus* and *P. balcanicus* (according to *Dufresnes et al., 2019a*, *2019b*). During the course of the present study, we registered only adults of the species *P. balcanicus*. No *P. fuscus* were found. On 28.06 (3) and 17.07.2020 (1), we were able to detect a total of four active specimens in the twilight after the sunset, but before the onset of complete darkness. On all of our working evenings, we spent 2h at the beginning of the night in an attempt to register the temperatures of the toads shortly after they became active. We were able to collect data from a total of 204 individuals. Each detected toad was immediately photographed by using "Panasonic Lumix FZ 200" (Panasonic Corporation, Kadoma, Osaka, Japan) and "Sony RX 10 III" (Sony Electronics Corporation, Minato, Tokyo, Japan). On site, we measured the temperature of the substrate at a depth of 11-12 cm by using "Ebro TLC 720" (Xylem Analytics Germany GmbH & Co. KG; EBRO, Ingolstadt, Germany). This device is equipped with a solid and sharp metal spike (with a length of 11.5 cm) on which is located a highly sensitive contact probe thermo-couple with accuracy of ±0.8 °C in the range of −18.0 to +119.9 °C.
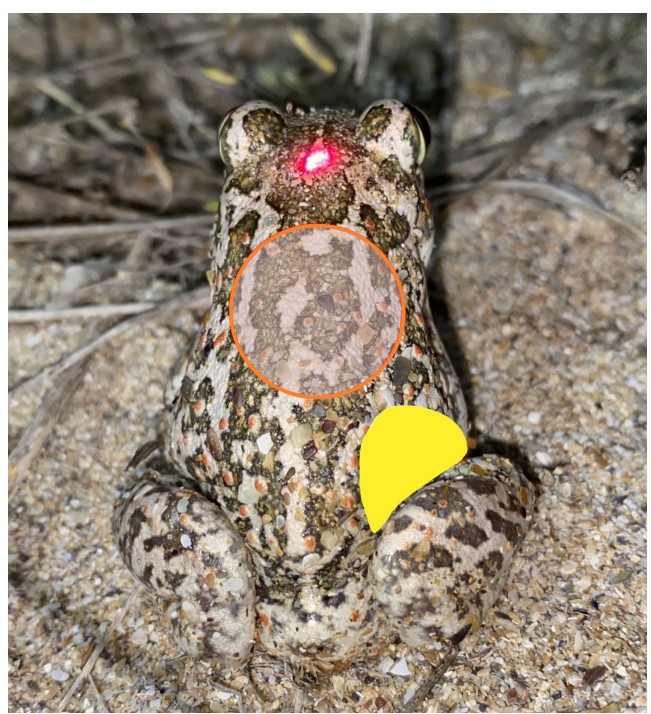

**Figure 1 Dorsal view of a specimen of *Pelobates balcanicus* during the procedure of thermal measurements.** The laser pointer of the IRT is aimed at the head of the toad; the red circle indicates the spot of measurement with the IRT; the yellow section represents the area where the temperature was measured by the use of the beaded thermocouple sensor.

For measurement of the body temperature of the spadefoot toads, we used a combination of several devices (see *Navas et al., 2013*). First, we measured the body temperature by using the Infrared measurement unit of "Ebro TLC 720" (further referred to as IRT). We used the factory set emissivity of 0.95. The device emits an IR beam with ratio Distance/Spot diameter = 5/1. According to the producer, the accuracy of the thermometer is ±2% in the range of −33 to +220 °C. The unit was oriented longitudinally to the midline of the toad as the laser pointer was aimed between the eyes (see Fig. 1) and the IRT beam was projected on the dorsal surface of the specimen (similar to *Mitchell & Bergmann, 2016*). At a distance of 125 mm between the beam source and the animal, the diameter of the measurement circle on the skin of the toad was 25 mm (see Fig. 1).

Additional measurements of the skin surface temperature were performed by using a Thermometer "Therma Elite 221-061" (ETI Ltd, Easting Close, Worthing, West Sussex, UK). The accuracy of the unit is ±0.4 °C, or ±0.1% in the range of −99.9 to 299.9 °C. As a K-thermocouple for the Thermometer, we used a "Fluke Electronics 80PK-1 K-Type Thermocouple Bead Probe" (Fluke Corporation, Everett, WA, USA). The combination of these two devices allowed for extremely fast temperature measurements of under 3 s. In accordance with previous protocols (see *Navas et al., 2013*), the temperature was measured with the beaded probe positioned in the inguinal area of the body (Fig. 1). We used this set also to measure the air temperature 12 cm above the substrate. Special care was taken to eliminate any contact between the investigator and the toads in order
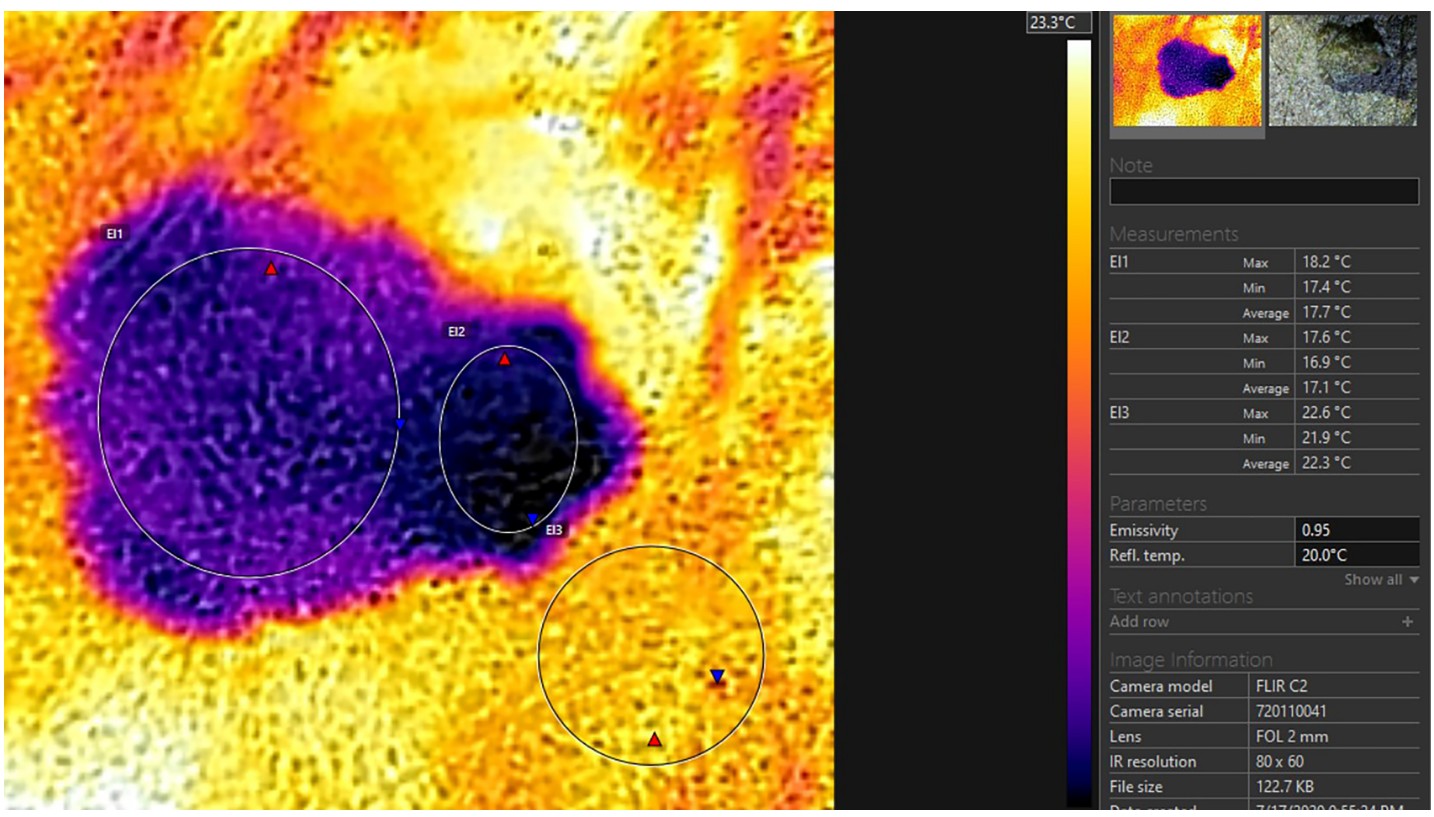

**Figure 2 Thermoprofile of a *Pelobates balcanicus* specimen represented by the "Flir tools" software.** The circles on the body (EI1) and on the head (EI2) represent the surface temperatures within these sections of the toad; the circle EI3 represents the surface temperatures of the substrate at the direct vicinity of the toad.

to avoid any thermal exchange (see *Navas & Araujo, 2000*). The only contact with the toads was performed by the tip of the K-Probe. In cases where the toads moved, they were immobilized by gently pressing them with a piece of plastic net against the substrate. All experiments were conducted in accordance with national animal welfare regulations-Permit number 767/21.01.2019 of the Ministry of Environment and Water, Bulgaria.

The thermal profiles (thermal images on which every pixel is loaded with thermal value) of the toads in dorsal projection and the surface of the substrate were recorded by the use of a thermal camera "FLIR C2" with an MSX Thermosystem (FLIR® Systems, Inc., Wilsonville, OR, USA). The emissivity settings were adjusted for the measurement of the frog's body temperature (according to *Rowley & Alford, 2007*). The accuracy of the system is ±1.5 °C. The radiometric images were analysed by the use of "FLIR Tools 6.X" software (FLIR® Systems, Inc., Wilsonville, OR, USA). The images were used for the calculation of the minimal, maximal, and average temperatures of the substrate surface in the closest possible proximity to the toad. The minimum, maximum and average temperatures of the bodies and heads were recorded (see Fig. 2). In one of our measurements, the toad defecated immediately before the obtainment of the thermal

**Table 1 Mean values of the temperatures of the soil, air and different areas of *Pelobates balcanicus* sorted by the dates of observation.**

| Date | t (deep) | t (air) | t (substrate) | t (TC) | t (IRT) | t (aver. body) | t (aver. head) | n |
|------|----------|---------|---------------|--------|---------|----------------|----------------|---|
| 14.6.2020 | 23.39 | 18.02 | 16.90 | 16.27 | 15.82 | 15.11 | 15.18 | 39 |
| 25.6.2020 | 26.07 | 22.55 | 21.57 | 20.94 | 20.37 | 20.06 | 20.05 | 42 |
| 27.6.2020 | 28.38 | 19.60 | 20.67 | 19.03 | 17.99 | 17.40 | 17.32 | 41 |
| 28.6.2020 | 29.15 | 20.60 | 22.39 | 18.90 | 17.63 | 17.80 | 17.49 | 60 |
| 17.7.2020 | 30.62 | 24.60 | 22.59 | 19.31 | 18.82 | 17.55 | 17.24 | 22 |

Note:

t (deep), temperature of the substrate at 12 cm depth; t (air), temperature of the air lair at the substrate surface; t (substrate), temperature of the substrate surface; t (TC), temperature measured by the use of the beaded thermocouple sensor in the inguinal section of the body; t (IRT), temperature measured by the use of IRT at the body surface of the toad; t (aver. body), average temperature within the circle EI1 (see Fig. 2); t (aver. head), average temperature measured in the circle EI2 (see Fig. 2); n, number of animals.

profile. In this case we additionally measured the temperature of the excrements, to compare it to the other measurements gained from the dorsal profile of the toad.

## Statistical analysis

The measurements of all parameters were analyzed by descriptive statistics (see Table 3). We used a Paired Two Sample for Means t-test on our data set as it meets the assumptions of the test. Several measurements in different body sections, as well as the environmental temperatures for each individual were compared. The confidence interval was calculated as a product of the statistical error and the critical value of the statistics t ($\bar{x} \pm t_\alpha \frac{\sigma}{\sqrt{n}}$) at a significance level of 0.05. The two mean values, which showed no statistical significance and are assumed to represent the null hypothesis were marked with an asterisk.

## RESULTS

Prior to the first evening of the investigation (14.6.2020), the weather was rainy and the substrate was rather wet, both on the surface, as well as in-depth resulting in the lowest temperatures measured for the substrate, but also for the 39 measured animals (see Table 3). In contrast, during the other visits, the weather conditions were dry. The highest number of the Balkan spadefoot toads with a total of 60 investigated animals was registered on 28.6.2020. The lowest number of toads observed on the last evening (17.7.2020) of our study with a total of 22 active animals. On 17.7.2020 we registered the highest temperatures of the air (24.6 °C) and also the highest average maximal temperatures of the substrate surface (22.59 °C). However, the average body surface temperature of the measured toads was relatively low (Table 3).

As represented in Table 1, the average temperature of the substrate in depth was relatively high, but on the surface, the maximal and minimal temperatures were in average about 9 °C lower (20.86 and 18.42 °C accordingly). The difference in temperature measurements by TC and IRT was on average 0.84 °C, while the respective difference in Thermal camera images and IRT was 0.53 °C. No statistically significant differences in the data provided by the measurements by the TC and the Thermal camera were found-the

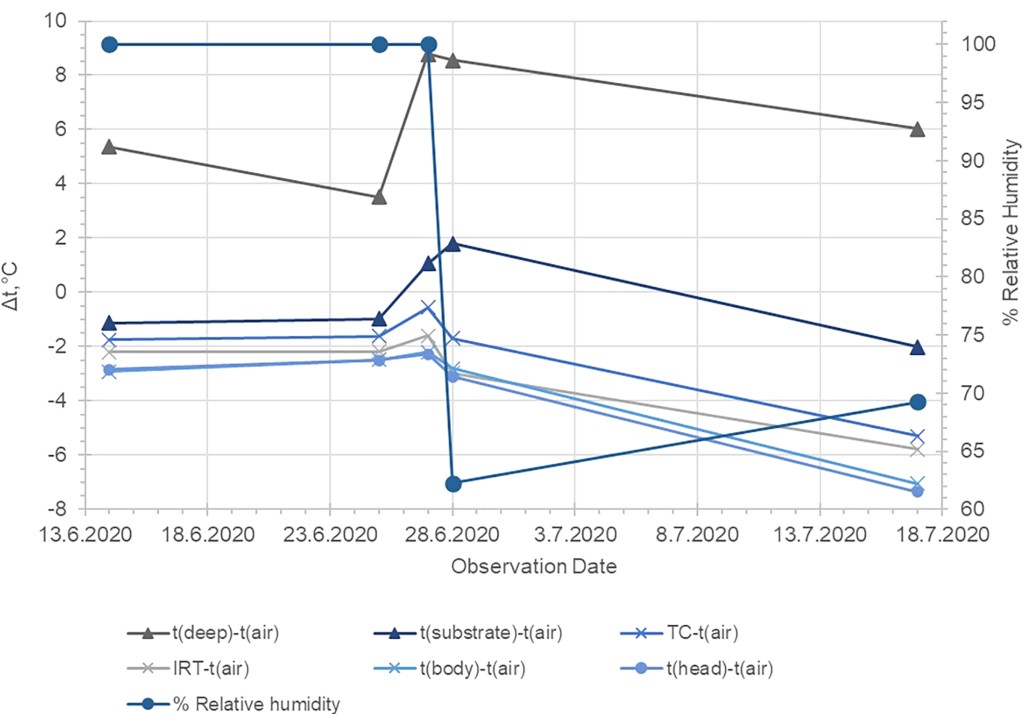

**Figure 3 Average temperature values in depth of the soil, on the soil surface, the ground air layer and in the different parts of the surface of the *Pelobates balcanicus*.** t (deep), temperature of the substrate at 12 cm depth; t (air), temperature of the air lair at the substrate surface; t (substrate), temperature of the substrate surface; t (TC), temperature measured by the use of the beaded thermocouple sensor in the inguinal section of the body; t (IRT), temperature measured by the use of IRT at the body surface of the toad; t (aver. body), average temperature within the circle EI1 (see Fig. 2) ; t (aver. head), average temperature measured in the circle EI2 (see Fig. 2).   

values were on the average of 0.31 °C (Table 1; Figs. 3–5). The average body temperature exceeded the average substrate surface temperature only in the first, second and fourth animals observed on 28.06 and in the first animal on 17.07. In all other cases, the temperature values measured in the different sections of the dorsal skin of *P. balcanicus* were lower than the average substrate temperature. The differences ranged from 0.6 °C to about 7.4 °C. All temperature measures taken in different areas of the dorsum of these toads were lower than air temperature (Fig. 4). During the first two evenings, the temperature differences in all measurements, except the deep substrate temperature, were relatively small (between 1 and 3 degrees). After these two initial evenings, the surface temperatures of the substrate deviated significantly from the measured temperatures from the animal's surface (Table 2). Substrate surface temperature was about 4–5 degrees than the measured body temperatures and even exceeded air temperature on two of the nights. Body temperature differences remained close in all nights of observation (Fig. 5).

  The temperature values of the different sections of the toads' dorsal profile had a dynamic character. In animals which were observed early after the start of our work (presumably recently emerging from the substrate), the temperature was comparatively higher and was closer to that of the subsoil on the spot. Shortly after that, body temperature presumably decreased rapidly and continued to change slowly, in correlation

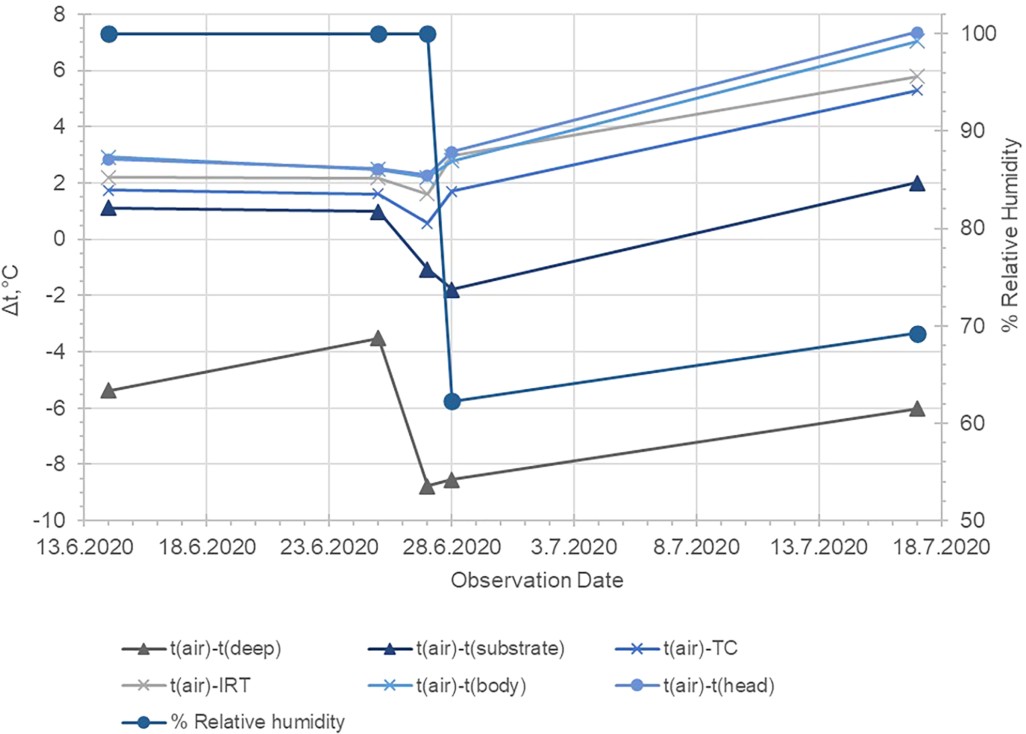

**Figure 4 Average temperature differences between the measured temperatures of the air, the substrate surface, the deep substrate and on the body and head surface of *Pelobates balcanicus*.** t (deep), temperature of the substrate at 12 cm depth; t (air), temperature of the air lair at the substrate surface; t (substrate), temperature of the substrate surface; t (TC), temperature measured by the use of the beaded thermocouple sensor in the inguinal section of the body; t (IRT), temperature measured by the use of IRT at the body surface of the toad; t (aver. body), average temperature within the circle EI1 (see Fig. 2); t (aver. head), average temperature measured in the circle EI2 (see Fig. 2). The relative humidity of the air is provided according to the website "Reliable Prognosis", from Weather station "Mangalia", Romania, WMO ID = 15499 (http://rp5.ru/archive.php?wmo_id=15499&lang=en).

with air temperature. Exponential decrease in temperature as a function of air temperature could be observed in all measured areas of the toads' bodies.

## DISCUSSION

Our results showed that the measured body temperatures of the toads were found to be below these of the surrounding air and the substrate surface in excess of 98% of specimens measured. However, the differences were within the range of 2–3 °C. In frogs, a strong correlation between the substrate surface and body temperature has been reported (*Carvajalino-Fernández et al., 2011*; *Gómez-Hoyos, Gil-Fernández & Escobar-Lasso, 2016*). However, these authors did not use methods that allow for discrete measurements of maximal and minimal temperatures. In most model studies, the aim was to define core temperature or body temperature, which can be included in the energy balance (see *Sousa et al., 2010*). However, careful examination of our thermograms showed that the body in *P. balcanicus* cools unevenly and on the dorsal surface there is a thermal gradient (Fig. 2). In our analysis, the temperature values obtained by the IRT appear to be

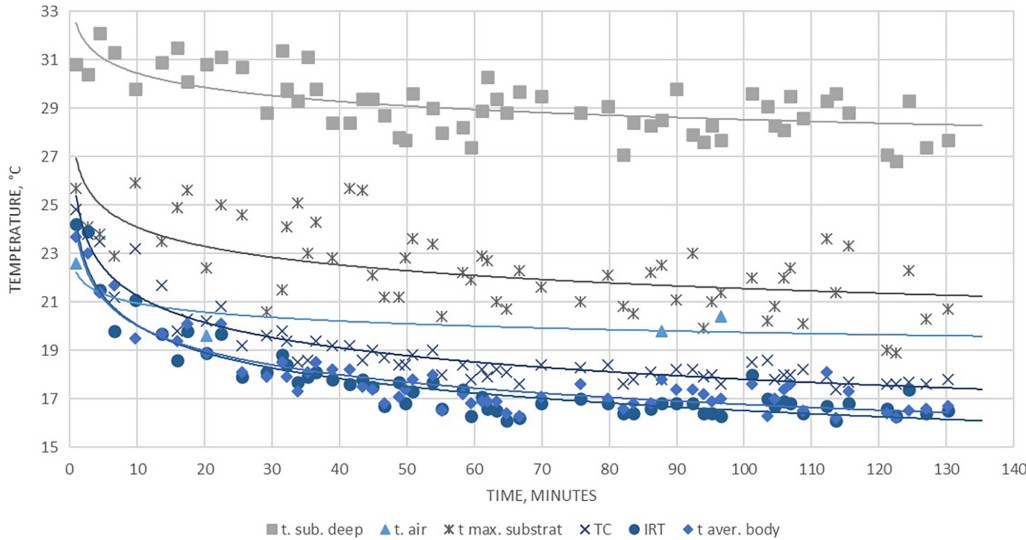

**Figure 5 Kinetics in the measured temperature changes of the substrate, the air and the different section of the surface of *Pelobates balcanicus* measured on 28.06.2020.** t (sub. deep), temperature at 12 cm under the substrate; t (air), temperature of the air; t (max. substrate), maximal temperature of the surface of the substrate; t (TC), temperature of the toad surface measured by the use of the beaded thermocouple sensor; t (IRT), temperature of the toad measured by the use of IRT; t (aver. body), average temperature of the body measured by the use of the thermal camera.

**Table 2 Average temperature differences between the different parts of the body of *Pelobates balcanicus* obtained from measurements with different instruments sorted by the dates of observation.**

| Date | t (TC) – t (IRT) | t (TC) – t (aver. body) | t (TC) – t (aver. head) | t (aver. body) – t (aver. head) | t (IRT) – t (aver. body) |
|---|---|---|---|---|---|
| 14.6.2020 | 0.456 ± 0.053 | 1.167 ± 0.079 | 1.090 ± 0.130 | −0.077 ± 0.086 | 0.710 ± 0.068 |
| 25.6.2020 | 0.564 ± 0.066 | 0.876 ± 0.273 | 0.886 ± 0.275 | 0.010 ± 0.028 | 0.312 ± 0.291 |
| 27.6.2020 | 1.041 ± 0.134 | 1.634 ± 0.227 | 1.715 ± 0.228 | 0.080 ± 0.046 | 0.593 ± 0.218 |
| 28.6.2020 | 1.268 ± 0.109 | 1.098 ± 0.159 | 1.408 ± 0.176 | 0.310 ± 0.070 | −0.170 ± 0.143 |
| 17.7.2020 | 0.491 ± 0.175 | 1.759 ± 0.292 | 2.068 ± 0.346 | 0.309 ± 0.130 | 1.268 ± 0.310 |

Note:
t (TC), temperature measured by the use of the beaded thermocouple sensor in the inguinal section of the body; t (IRT), temperature measured by the use of IRT at the body surface of the toad; t (aver. body), average temperature within the circle EI1 (see Fig. 2); t (aver. head), average temperature measured in the circle EI2 (see Fig. 2).

consistently lower than those measured by the use of the thermocouple system. For both measurements, we used high-quality instruments and both methods were used largely in temperature investigations in anurans (for an overview see *Khozatskii, 1959*, *Rowley & Alford, 2007*). We propose that the differences in the measured values are related to the section of the body surface from which the thermal information was obtained. The coldest spot in the dorsal thermal profile of *P. balcanicus* was always the head. On the dorsal body surface, we detected that the positive temperature gradient is increasing (the differences may reach over 1.5 °C) in the direction of the posterior-lateral body sections. We obtained the temperature data from the TC from the inguinal section and this may

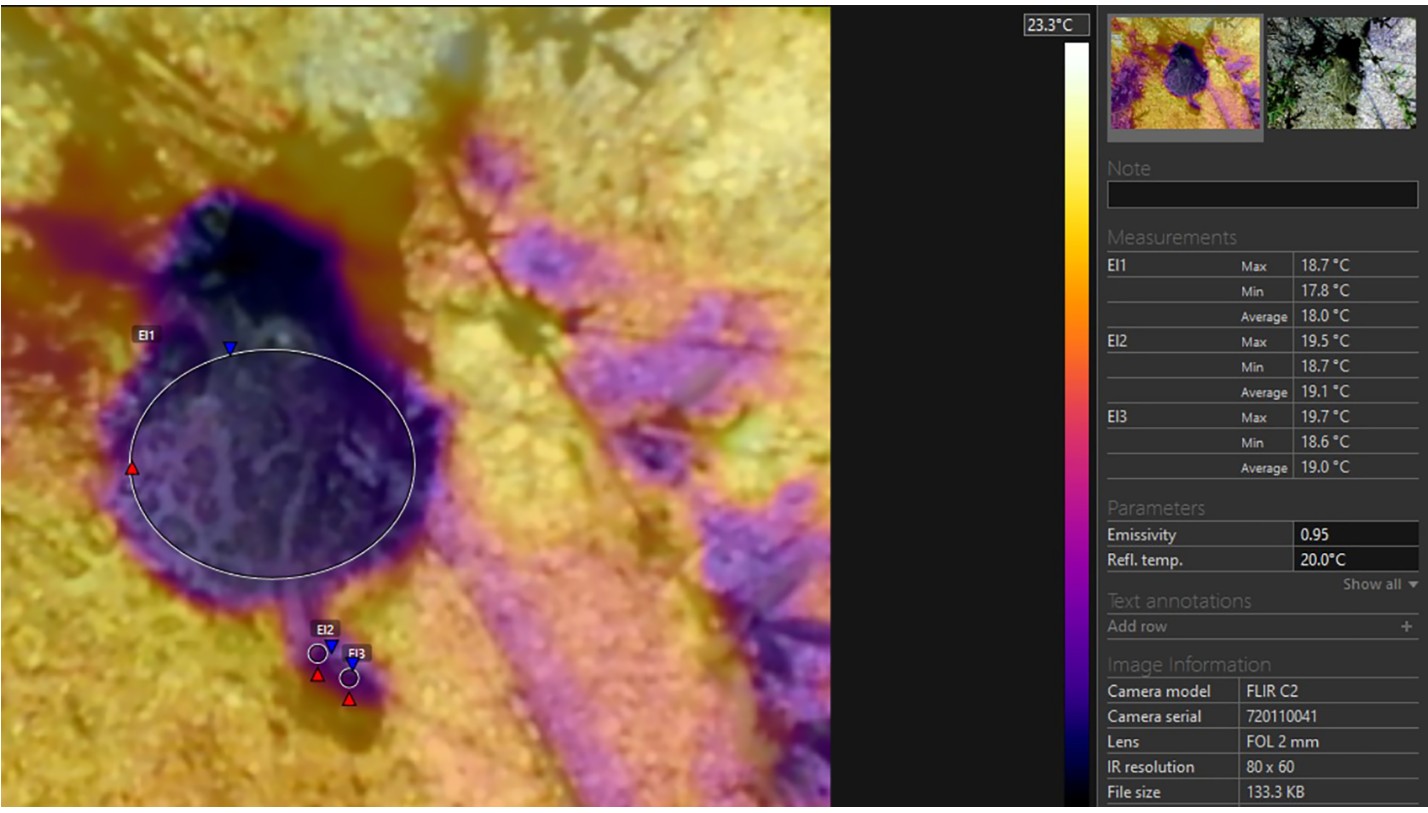

**Figure 6 Thermal image of a specimen of *Pelobates balcanicus* during defecation.** EI1, maximal, minimal and average temperature of the body of the toad; EI2, maximal, minimal and average temperature of the proximal section of the excrement; EI3, maximal, minimal and average temperature of the distal section of the excrement.       

explain the higher values. That practically means, that the instruments worked precisely and delivered reliable measurements, which reflect the actual distribution of the temperatures on the dorsal skin.

The thermal profile represented in Fig. 6 demonstrates that the temperature of the body and the temperature of the excrements during defecation in *P. balcanicus* were within the range of 1.1 °C. For small-sized amphibians, *Rowley & Alford (2007)* report differences between skin surface temperature and cloacal temperature in the range of 0.5 °C. The combined use of measurement instruments used in the present study demonstrated that the data collected by all three systems are reliable and all differences in the values are consistent. The IRT, the penetration probe thermocouple, the K-type beaded thermocouple thermometer and the thermal camera complement each other and their parallel use allows for a fast acquisition of large amounts of thermal data, both from the biological objects, as well as the substrate of their microhabitat. The measurements were performed in few seconds and do not require any contact between the researcher and the anurans. The field methods we applied were non-invasive, and should be considered for similar studies as it significantly reduces the need to capture and handle small sized ectotherm tetrapods.

The mechanism which allows the Balkan spadefoot to remain slightly cooler compared to the surroundings cannot be fully understood by direct measurements. Under controlled conditions, the processes of thermoregulation in amphibians are well studied, but data from field investigations are rather scarce (see *Feder, 1992*). Most amphibians possess a very low potential for thermoregulation (*Brattstrom, 1979*). According to *Mokhatla, Measey & Smit (2019)*, both thermoregulation and metabolic rate are not only related, but fully controlled by environmental factors. In the framework of the thermal inertia concept, *Carey (1978)* demonstrated that larger toads cool down more slowly than smaller specimens. According to *Tracy et al. (1993)*, however, the most relevant factors were considered to be temperature, body water balance and water evaporation regulation. During hot days, for most amphibians it is critical to keep body temperature below 40 °C (*Brattstrom, 1963*, *1968*). They may search for cooler spots by means of migratory behavior (*Young et al., 2005*), or may rely on cutaneous water evaporation to regulate their temperature (*Duellman & Trueb, 1986*; *Mitchell & Bergmann, 2016*). This fits the first scenario proposed by *Spotila, O'Connor & Bakken (1992)* for the impact of the surrounding factors on the anurans' temperature. In this case, the sun will directly warm the body, causing evaporation by skin and will precipitate the effect of water evaporation cooling. From a thermo-physical perspective, the cooling effect of water evaporation from the body surface may be crucial for the amphibians to maintain low temperatures during sun exposure (*Brattstrom, 1963*). However, intensive water evaporation may lead to water imbalance in frogs. Body hydro-regulation is even more challenging than thermoregulation in some amphibians (see *Lillywhite & Navas, 2006*; *Mitchell & Bergmann, 2016*). For two aquatic and one terrestrial anurans, *Mokhatla, Measey & Smit (2019)* reported that at high air temperatures, the body temperature remained lower than the surrounding air. The authors report differences of 4–7 °C and propose that all three species rely on evaporative cooling at higher temperatures.

According to *Spotila, O'Connor & Bakken (1992)*, under nocturnal conditions, the humidity contained in the air could condense on the surface of the toad forming a "dew layer" and simultaneously (if in the open) the toad could also lose heat against the sky (especially if the sky is clear). These authors stressed that such phenomena were not an object of direct investigation. However, such a scenario would only be possible in the event of a dramatic difference in the temperature of the air and the anurans.

In all of our observations, the highest temperature was measured deep within the substrate. It exceeded the air temperature by 3.5 to 8.8 degrees. In three of the nights, the soil surface temperature was lower, and in two it was close (1–2 degrees higher) to the air temperature. After sunset, any surface can only be warmed by ground heat and cooled by air exchange and (mostly) by evaporation (*Spotila, O'Connor & Bakken, 1992*). Surface temperature should depend on both the air humidity and the humidity within the soil. According to *Eggert (2002)*, spadefoot toads spend several days in a row buried without any activity at night-in our case in warm and moist sandy soil. Actually, after sunset, only part of the toads appears on the surface and this might be related to thermoregulation and

water balance regulation. At the beginning of the night, the body temperature is determined by the heat and moisture accumulated from the soil. The body temperature is practically independent from absorption of solar radiation (see also the conclusions of *Tracy, 1976*). Above ground, the toads engage in heat exchange with the air and the surface of the substrate through heat conduction and convection (see *Spotila, O'Connor & Bakken, 1992*). These processes can lower the body temperature to equalize it with the ambient temperature. The further decrease in body temperature is due to the evaporation of a part of the absorbed moisture. We propose that the described heat exchange processes take place quickly (presumably in the initial 10–20 min of the night activity) and the evaporation rate is low. In the following stages, the changes are small and synchronised with the air temperature variation-approximate thermal equilibrium is reached. Unlike water frogs, the skin of *Pelobates* is not covered with a fluid layer (*Stojanov, Tzankov & Naumov, 2011*). It has a non-uniform structure-in some sections, it is rough and warty (with a large surface for evaporation), and in other sections, it is smoother (and perhaps with higher permeability). We propose that this inhomogeneity causes uneven surface humidity and due to uneven evaporation, temperature differences occur, which contribute to body heat gradients. These are visible in the thermal images (Figs. 2 and 6). Thermal gradients on the skin surface were reported previously in *Rana temporaria, Bufo bufo* and *Bufotes viridis* (*Khozatskii, 1959*). The data of Khozatskii were obtained by the use of contact thermocouple thermometer and cannot be compared directly to the data delivered by our thermal camera, however the gradient patterns resemble these detected in *P. balcanicus. Khozatskii (1959)* explains the thermal gradient in anurans by differences in the distribution of the subcutaneous blood vessels involved in the cutaneous gas exchange. The author proposed that local differences in the water evaporation intensity on the skin surface in *R. temporaria* also contribute to the thermal gradient.

Our results indicate that the temperature differences between the environment and the toads' dorsal surface correlated to the change in the air humidity. Based on meteorological data from the nearest weather stations (Weather station WMO ID = 15499: link under Fig. 4), we added relevant graphs of relative humidity in Figs. 3 and 4. During the first three nights of our investigation, the air humidity was 100% and this might have reduced the possibility of evaporation from the skin of the animals (low water loss). As the relative humidity lowers, the temperature differences between the toads' bodies and the substrate surface increased (presumably because the water evaporation through the skin increased). The spontaneous evaporation from the soil surface depends not only on the air humidity, but also on the soil moisture which may explain that the surface temperature of the soil even exceeded that of the air on June 27 and 28. On those nights, the soil was presumably drier, so the heat absorbed during the day was released more slowly due to the lack of evaporation from the soil. On the basis of those analyses we can propose that the potential water loss in hotter and dryer nights may be one of the crucial factors which constrain the activity of the investigated toads. Our work indicates such a trend. However, the data are too limited to allow for definite conclusions. The potential relation between the environmental conditions and the behaviour in pelobatids has to be proven by field

**Table 3 Temperatures of the soil, air and different areas of *Pelobates balcanicus*.**

| Date: 14.6.2020 | t (deep) | t (air) | t (IRT) | t (TC) | t (max. body) | t (min. body) | t (aver. body) | t (max. head) | t (min. head) | t (aver. head) | t (max. substrate) | t (min. substrate) |
|---|---|---|---|---|---|---|---|---|---|---|---|---|
| Mean | 23.39 | 18.20 | 15.82 | 16.27 | 15.36 | 14.94 | 15.11 | 15.35 | 15.01 | 15.18 | 16.90 | 15.89 |
| Standard deviation | 0.95 | 0.00 | 0.95 | 0.95 | 0.95 | 0.91 | 0.92 | 0.91 | 0.94 | 0.95 | 1.05 | 0.92 |
| Minimum | 21.3 | 18.2 | 13.3 | 13.7 | 12.9 | 12.6 | 12.9 | 13 | 12.5 | 12.7 | 14.3 | 13.6 |
| Maximum | 25.9 | 18.2 | 17.4 | 17.8 | 16.8 | 16.4 | 16.5 | 16.8 | 16.5 | 16.8 | 18.9 | 17.3 |
| (n) | 39 | 39 | 39 | 39 | 39 | 39 | 39 | 39 | 39 | 39 | 39 | 39 |
| date: 25.6.2020 | | | | | | | | | | | | |
| Mean | 26.07 | 22.79 | 20.37 | 20.94 | 20.31 | 19.88 | 20.06 | 20.27 | 19.88 | 20.05 | 21.64 | 20.02 |
| Standard deviation | 0.62 | 0.08 | 0.47 | 0.39 | 0.69 | 0.69 | 0.68 | 0.70 | 0.67 | 0.68 | 1.23 | 0.84 |
| Minimum | 24.4 | 22.3 | 19.9 | 20.1 | 19 | 18.5 | 18.8 | 18.9 | 18.7 | 18.8 | 19.6 | 18.6 |
| Maximum | 27.3 | 22.8 | 21.8 | 21.9 | 21.5 | 21 | 21.2 | 21.5 | 21 | 21.1 | 24.2 | 21.8 |
| (n) | 42 | 42 | 42 | 42 | 42 | 42 | 42 | 42 | 42 | 42 | 42 | 42 |
| date: 27.6.2020 | | | | | | | | | | | | |
| Mean | 28.38 | 19.69 | 17.99 | 19.03 | 18.37 | 17.09 | 17.40 | 18.11 | 17.06 | 17.32 | 20.67 | 17.87 |
| Standard deviation | 1.63 | 0.45 | 0.79 | 0.77 | 1.19 | 1.05 | 1.06 | 1.06 | 1.00 | 1.02 | 1.51 | 1.13 |
| Minimum | 24.6 | 18.8 | 16.4 | 17.4 | 16 | 15.3 | 15.7 | 16.1 | 15.4 | 15.7 | 17 | 15.6 |
| Maximum | 32.8 | 20.3 | 19.4 | 20.8 | 20.8 | 19.7 | 20.1 | 20.3 | 19.5 | 19.8 | 24.1 | 20.7 |
| (n) | 41 | 41 | 41 | 41 | 41 | 41 | 41 | 41 | 41 | 41 | 41 | 41 |
| date: 28.6.2020 | | | | | | | | | | | | |
| Mean | 29.15 | 20.22 | 17.63 | 18.90 | 19.43 | 17.34 | 17.80 | 18.65 | 17.07 | 17.49 | 22.39 | 19.06 |
| Standard deviation | 1.24 | 0.10 | 1.70 | 1.62 | 1.41 | 1.52 | 1.58 | 1.47 | 1.28 | 1.43 | 1.74 | 1.04 |
| Minimum | 26.8 | 19.6 | 16.1 | 17.4 | 17.1 | 15.5 | 16.2 | 16.5 | 15.5 | 15.9 | 18.9 | 17.2 |
| Maximum | 32.1 | 22.6 | 24.2 | 24.8 | 24.2 | 22.9 | 23.7 | 23.6 | 22 | 23.1 | 25.9 | 22 |
| (n) | 60 | 60 | 60 | 60 | 60 | 60 | 60 | 60 | 60 | 60 | 60 | 60 |
| date: 17.7.2020 | | | | | | | | | | | | |
| Mean | 30.62 | 24.62 | 18.82 | 19.31 | 19.07 | 17.15 | 17.55 | 18.02 | 16.92 | 17.24 | 22.59 | 19.11 |
| Standard deviation | 1.04 | 0.16 | 1.64 | 1.64 | 1.41 | 1.26 | 1.35 | 1.25 | 1.04 | 1.13 | 1.49 | 1.12 |
| Minimum | 28.8 | 24.4 | 16.6 | 17.2 | 16.8 | 15.4 | 15.8 | 16.2 | 15.4 | 15.6 | 19.9 | 17 |
| Maximum | 32.4 | 24.8 | 23.8 | 24.2 | 23.4 | 21.5 | 22.2 | 21.3 | 20.1 | 20.8 | 25.1 | 21 |
| (n) | 22 | 22 | 22 | 22 | 22 | 22 | 22 | 22 | 22 | 22 | 22 | 22 |

**Note:**
t (deep), temperature of the substrate at 12 cm depth; t (air), temperature of the air lair at the substrate surface; t (IRT), temperature measured by the use of IRT at the body surface of the toad ; t (TC), temperature measured by the use of the beaded thermocouple sensor in the inguinal section of the body; t (max. body), maximal temperature of the body of the toad; t (min. body), minimal temperature of the body of the toad; t (aver. body), average temperature of the body of the toad; t (max. head), maximal temperature of the head of the toad; t (min. head), minimal temperature of the head of the toad; t (aver. head), average temperature measured of the head of the toad; t (max. substrate), maximal temperature of the surface of the substrate; t (min. substrate), minimal temperature of the surface of the substrate; n, number of animals.

investigation and in the present study we propose a theoretical frame which can be experimentally tested in the future.

## ACKNOWLEDGEMENTS

Francisco Javier Zamora-Camacho, Alexander Westerström and an anonymous reviewer are acknowledged for their helpful comments and the improvement of the quality of the manuscript. AW performed a proof read of the manuscript and improved the English version.

### Funding

This work was supported by the Bulgarian Ministry of Education and Science, Grant No. RD-08-67/25.01.2021. The work of Daniel Jablonski was supported by the Slovak Research and Development Agency under contract No. APVV-19-0076. The funders had no role in study design, data collection and analysis, decision to publish, or preparation of the manuscript.

### Grant Disclosures

The following grant information was disclosed by the authors:
Bulgarian Ministry of Education and Science: RD-08-67/25.01.2021.
Slovak Research and Development: APVV-19-0076.

### Competing Interests

The authors declare that they have no competing interests.

### Author Contributions

- Nikolay Natchev conceived and designed the experiments, performed the experiments, analyzed the data, prepared figures and/or tables, authored or reviewed drafts of the article, and approved the final draft.
- Teodora Koynova conceived and designed the experiments, performed the experiments, analyzed the data, prepared figures and/or tables, authored or reviewed drafts of the article, and approved the final draft.
- Krasimir Tachev performed the experiments, analyzed the data, prepared figures and/or tables, authored or reviewed drafts of the article, and approved the final draft.
- Dimitar Doichev performed the experiments, prepared figures and/or tables, and approved the final draft.
- Pavlina Marinova performed the experiments, prepared figures and/or tables, and approved the final draft.
- Valeriya Velkova performed the experiments, prepared figures and/or tables, and approved the final draft.
- Daniel Jablonski conceived and designed the experiments, performed the experiments, analyzed the data, authored or reviewed drafts of the article, and approved the final draft.

### Animal Ethics

The following information was supplied relating to ethical approvals (*i.e.*, approving body and any reference numbers):

Permit number 767/21.01.2019 of the Ministry of Environment and Water, Bulgaria. All experiments were conducted in accordance with national animal welfare regulations.

### Data Availability

The raw data is available in the Supplemental Files.

## Supplemental Information

Supplemental information for this article can be found online at http://dx.doi.org/10.7717/peerj.13647#supplemental-information.

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
