# Peer review of "Temperature regulation in the Balkan spadefoot (Pelobates balcanicus Karaman, 1928) at the beginning of nocturnal activity"

_PeerJ, doi:10.7717/peerj.13647_

## Round 0.1 · original submission · Major Revisions

Two recognized experts have assessed your manuscript and identified a number of issues that the manuscript revision should resolve. In order to be accepted, the manuscript needs a comprehensive and far-reaching revision. Please consider the proposed changes as mandatory.

I look forward to your revised manuscript.

Reviewer 1 ·

Basic reporting

Overall the ms is concise and well written and presented. However, it has (few) linguistic issues in certain parts which I (not an native English speaker) have tried to ameliorate.
I present these issues along with one issue in the methodology in the line by line comments below. I recommend the article be published after a minor revision to address the specific points mention.

Line 100: P. syriacus balcanicus.
Lines 100-102: “…we were able to register and analyze only … P. balcanicus”. Does this mean that the authors did not locate P. fuscus? Or did not discriminate between the two? My guess is the first, but if yes, this should be more clear.
Lines 102-103: “On 25.06; 28.06 and 17.07.2020 we were able to detect a total of five active specimens in the dusk period before the full darkness.” I, again, guess that the authors here want to stress that the animal does not become active before full darkness. If this is their point, it is not clear. Please rephrase. Moreover, please add numbers after the dates e.g. something like 25.06 (2) 28.06 (1) …
Lines 160-162: The syntax here is problematic. Maybe change “were” to “we”, or rephrase.
Line 180: exceeds > exceeded
Line 184: have > were
Lines 184-185: here and further down (lines 234-236) the authors mention thermal differences regarding animals which had emerged before a small time period. Nevertheless, they do not mention how they did understand that the animals had just emerged. There is a point where they mention animals covered with sand as an indicator. But I think the authors should make a clear statement about this, even if is only a qualitative approach (e.g. “covered with sand”)
Line 185: Soon later the body… > Soon after, the body…
Line 186: decreased rapidly and continues > decreased rapidly and continued
Lines 186-189: “decrease in temperature could be observed as a function of time in all measured” should read “decrease in temperature as a function of time could be observed in all measured”. Moreover, the meaning of the second part of the phrase “while the soil samples at the same sites showed large dispersion and lower correlations.” is not clear to me. Please rephrase/clarify.
Lines 193-194: please consider the following wording: Most amphibians possess a very low potential for thermoregulation (Brattstrom, 1979).
Lines 198-199: please rephrase. Consider e.g.: Water temperature, body water balance and the way the animals deal with it by means of physics of vaporized water, are regarded as the most important factors (Tracy et al. 1993). Moreover, could the part “by means of physics of vaporized water” could be replaced by evaporation? Perspiration? Other relevant term? Finally, please recheck the connection of this phrase with the previous one.
Line 227: On certain nights, some toads start their activities on the surface. I do not understand the meaning of this phrase.
Lines 234-236: how did the authors determine that the animal they saw had emerged 10’, 20’ or 60’ ago? Please see also comment for lines 184-185. Maybe, time after sunset could be an indicator for what they want to demonstrate.
Line 247: With lowering the relative humidity > “with the lowering of relative…” or “as the relative humidity lowers…” or similar
Lines 279-280: except for the syntax problem, this phrase is too strong a statement. Please consider something like: “The field method we applied is noninvasive, and should be considered for similar studies as it dramatically reduces the need to capture and handle small sized ectotherm tetrapods”.

Experimental design

No comment

Validity of the findings

No comment

·

Basic reporting

This article tackles an interesting topic, which is thermoregulation in amphibian. These animals are generally considered thermoconformers, which does not mean that heat exchange processes do not necessarily play a central role in their ecology and physiology. Therefore, this piece of research could make a fine contribution to the area of thermal biology.

However, the current version of the manuscript needs more work. In the first place, the English needs a thorough revision. Below are some specific points where the grammar needs improvement. However, just addressing those points will not be enough to make the text readable. I strongly recommend the authors have their manuscript revised by a fluent speaker of English.

Moreover, the abstract needs rewriting. As it is, it goes back and forth between background and methods. The first sentences of the abstract should contextualize the whole study, stating why the topic is relevant and what is known. Then, it should briefly describe the goals of the study. Next, your methods should be explained. Finally, the main results should be presented and discussed. The reading will be much smoother by following this sequence. Moreover, I find the last sentence of the abstract kind of misleading, as, unlike this sentence suggests, this article does not look into the relationships between thermoregulation and behavioral shifts. I’d suggest the authors rewrite this section to give a clearer idea of what will be studied in the manuscript.

The introduction is better organized, but it still could benefit from some polishing. I recommend the authors rethink it, always bearing in mind what they did, what they want to tell, and how the introduction serves that purpose.

Experimental design

One of my main concerns with this article is that I did not get a clear view on what the aims of the authors were. Was it a comparison between the temperature in the environment (air and underground) and the toads? In the results section, it would seem that they wanted to compare sampling days, as if their objective was a description of the thermal relationships between toad and environment as the season progresses. But that is never stated in the methods section. And, if that was the aim, a more detailed description of the sampling protocol (sampling effort, how the sampling dates were chosen, etc.) is needed. Also, some sampling days were rainy whereas others were not. I wonder whether that would affect the willingness of toads to abandon their burrows, and the thermal relationships between the environment and the animals, which would make the results between days hardly comparable.

Moreover, the probes and devices used must be better described. Just stating the trademark won’t provide the reader with a sufficient idea of how it works. The reader needs to know the type of information they give, and why it was necessary to use different devices to obtain different measures. Was a comparison between techniques one of the goals of the study? It was never stated. Likewise, in the discussion (and even in one of the figures) the temperatures of feces are described. But details on how and why these measurements were taken are not sufficiently described in the materials and methods section. How many feces were measured? What is more, the total sample size, i.e., how many toads were measured, should be clearly stated. Why were sexual differences not assessed?

Also, the statistics are vaguely described. Also here, I get the impression that a greater clarity in the aims of the study would be reflected upon a better described statistics section. I would like to remind the authors that all statistical tests conducted should be clearly described here, in the same order in which the results will be presented.

I recommend the authors think clearly what their aims are, how their data contribute to those aims, what information they can give, and rewrite the materials and methods section explaining what they did and how they obtained that information.

Validity of the findings

More information is essential, especially concerning the characteristics of the devices used and why they were needed, so as to evaluate the validity of the results reported. Moreover, the results themselves should be presented in a clearer fashion. Many parts of the results sections appear to be too descriptive. My suggestion is that the authors stick to the statistical analyses performed (which are not sufficiently described, neither in the statistics nor in the results sections) and give the values (r, chi-square, F-statistics, P-values…) derived from each test. Again, comparisons among days are not necessary unless they are a part of the aims, which was never stated.
Similarly, many paragraphs of the conclusions appear unrelated to the findings of the authors, and function mostly as a description of amphibian heat exchange in different circumstances. I recommend a more direct comparison between the results obtained here and those by other authors.

Additional comments

In general, I find that this article is well-meaning, but needs more work to accomplish its goals. And that is, to my mind, its main weakness: its goals are not clear. I recommend the authors rethink what their real aims are, and rewrite the whole text to serve the purpose of clearly explaining those aims: why those aims, how they were reached, and what they mean in a wider context. Stay away of everything that’s merely anecdotal, and stick to the statistical results obtained and their biological significance. Writing a scientific article is an arduous task which requires a fair amount of thinking, most of which has to do with the structure of it. I recommend the authors devise clearly what is the “story” the want to tell, and structure the entire article bearing it in mind. Tell that story, and only that story, following these tips:

The introduction should make a pertinent summary of what is known about the subject, without giving superfluous information though. It should begin with general information, and become gradually specific. Particularly important is the segue: make sure your concepts and idea follow a cadence that is as smooth as possible. And, to conclude, state clearly what objectives this piece of research intends to accomplish.

The materials and methods section should describe the species, the study system, the sample sizes, the devices used (not only should the trademarks be stated, but also the kind of information they provide), and, in a separate section, the statistical analyses performed, clearly described.

The results section should, in an orderly fashion, provide sufficient statistical detail to let the reader know which effects were significant and which were not, following the sequence described in the statistics section.

The discussion should connect the novel findings described in the previous section with what is already known, stressing their significance. Ideally, the main results should be described first, and then gradually develop the secondary results. However, this is susceptible of change depending on the very nature of the data themselves.

Also, the English should be formal, advanced, and correct. Below is a list of mistakes that need amendment. However, I strongly recommend the text be thoroughly revised by a competent native speaker of English.

Line 18: what does “behavior mode” mean? If it just means “behavior”, I’d remove the word “mode”. And if it’s a valid concept, it needs clarification.
Line 22: I’d remove the adjective “bright”, as well as the word “the” before “sandy”. Plus, “substrate” should be in the plural form.
Line 23: remove the period after “cm”.
Line 25: a comma is needed after “measurements”. Also, the past tense should be used here.
Line 27: I’d say the average reader won’t know what a K-type thermocouple is. I suggest a brief explanation.
Line 28; “and a thermal camera…”.
Line 29: “basis”.
Line 36: “have changed”.
Lines 38-39: change “in the ectotherm organism” to “in ectotherms” or “in ectothermic organisms”.
Line 41: change “extreme activities level of their muscular system” to “extreme levels of muscle activity”.
Lines 39-41: this sentence fails to connect ectotherms in general to amphibians in particular, as it disregards other ectotherms which do perform at high body temperatures by actively thermoregulating. I suggest a milder segue, something like “Although most ectotherms invest very little in heat production, some thermoregulate actively, whereas others are thermoconformers. Such is the case of most amphibians”. You will easily find citations to support these statements.
Line 43: change “coasts” to “costs” (this mistake has been made twice in this sentence).
Line 44: change “, however small” to “. However, small”.
Line 45: “cold-blooded” (which is the correct spelling) is an old-fashioned concept that should no longer be used. I recommend using “ectotherms” instead.
Line 47: change “media” to “surroundings” or “environment”.
Line 49: “heat storage”.
Lines 49 and 51: change “would” to “should”.
Line 50: delete “the” before “air”.
Line 52: delete “the” before “metabolism”.
Line 54: change “or” to “and”.
Line 56: change “inhabited media” to “habitat” or “microhabitat”.
Line 56: change “exchanges” to “exchange”.
Line 57: delete “of” before “convection” and “The” before “temperature”.
Line 58: “thermodynamic”
Line 61: reword to “different types of sun rays”.
Line 64: “coast” should be changed to “cost”.
Line 67: a new paragraph should begin here.
Line 67: “In this paradigm” should be changed to “In this context”.
Line 68: add a comma after “species”.
Lines 68-69: why did you focus on the dusk period in the early summer? This sentence is more appropriate in the discussion section, where it’ll be clearer after explaining the biorhythms of this species.
Line 69: make clear what a thermal profile is. And add “and” before “the profile”.
Line 72: what is “the thermal specific”? Please, explain. I’m not familiar with this concept.
Line 74: add comma after “(1992)”, and change “had” to “have”.
Line 75: what are “elements” in this context?
Line 76: change “burrowing” to “burrows”.
Line 86: the capital “s” in “spadefoot” is incorrect, here and throughout the manuscript.
Line 99: change “concerning” to “pursuant to”.
Line 102: change “from” to “of”. And add comma after “2020”.
Line 105: change “spend” to “spent”.
Line 106: change “become” to “became”.
Line 125: delete “at” before “12 cm”.
Line 127: change “to” to “with”.
Line 128: change “willing” to “prone”.
Line 129-130: this statement isn’t even a proper sentence. I am not sure this is the appropriate location for it.
Line 144: what does “our values” mean in this context?
Lines 145-146: this sentence is too ambiguous. What measurements specifically? How were they compared? Also, tables should be cited in the results section, not here.
Line 149: this sentence is not properly written. Is the comma spare?
Lines 153-163: This entire paragraph is mostly anecdotal, and the information given here does not really contribute to the presentation of your scientific results. I would rethink this paragraph, or even delete it, unless you intended to present a work on the seasonal progression of toad activity and temperature. If that is the case, more information is needed, regarding how the sampling was done, what the sampling effort was… A problem to that, though, is that the weather was not similar in the different sampling nights, which could affect the activity of the toads and thus make it difficult to compare activity among nights.
Lines 153-154: this sentence should be deleted, as it does not add anything to the text.
Line 156: were these differences statistically significant? This should be clearly stated.
Line 159: change “were” to “was”.
Line 160: change “were” to “we”.
Line 162: the comma after “(22.59ºC)” should be a period.
Lines 166-168: this information was already given in the methods section. Thus, there is no need to repeat it here.
Line 170: a comma is needed after “camera”. Although this sentence could well be rewritten, as it does not sound natural the way it is.
Line 173: delete “the” before “air temperature”.
Lines 174-176: sentences like this are to be avoided. Instead, you should clearly state your results and indicate the figure where they can be seen, for instance: “All temperature measures taken in different areas of the dorsum of these toads were lower than air temperature (Fig. 4).”
Line 177: a comma is needed after “temperature”.
Line 178: please, be specific: higher or lower?
Line 179: delete “the” before “substrate”.
Line 180: “exceeded”. Also, delete “the” before “air temperature”.
Line 181: “remained”.
Line 184: “were”. Also, how do you know the time elapsed since the toads emerged?
Line 185: “spent”. Also, change “soon later the body…” to “shortly after that, body…”.
Line 186: “continued”. Also, delete “the” before “air temperature”.
Lines 183-189: please, provide the statistics that prove all these assertions.
Lines 193-194: rephrase into “For most amphibians, a limited thermoregulation potential has been stated”.
Line 196: “environmental factors”.
Line 197: “more slowly”.
Line 198: “The most relevant factors were considered temperature, body water balance…”
Line 201: This dash should be a period.
Line 209: This dash should be a period.
Lines 209-210: “Body hydro-regulation is even more challenging than thermoregulation in some amphibians”.
Line 230: please, discuss these conclusions here.
Line 237: This dash should be a period.
Line 247: “With dwindling relative humidity”.
Line 248: “the toads’ bodies”.
Lines 255 and 257: The comma before “however” should be a period.
Line 267: “thermal”.
Line 271: was this aim described in the methods section?
Line 280: “ectothermic tetrapods”.

---

## Round 0.2 · Minor Revisions

As also pointed out by the reviewer, despite some improvements, the manuscript still requires a fundamental linguistic revision. I therefore strongly recommend calling in an editorial service. The manuscript is good in terms of content and also with regard to the statistical validation of the findings, so that no further changes are necessary here, except for the restructuring of the abstract, as recommended by the reviewer.

·

Basic reporting

The new version of this manuscript has noticeably improved after the changes implemented. However, there are still multitudinous grammatical errors to be corrected. The authors claim that a native speaker of English reviewed the manuscript. Yet, the writing is still full of language mistakes that make the reading difficult and hard to follow. Below is a detailed list of mistakes I have spotted. Nonetheless, I strongly recommend the authors seek the aid of another fluent speaker of English who does a better job. As it is, the quality of the findings described is overshadowed by the poor writing.

Line 19: Not necessarily incorrect, but “conditions” would be more appropriate, as physiology is usually multi-faceted.
Line 21: Again, “changes” would be more adequate.
Line 21: “In the range of our study” does not add much information. I suggest replacing with the actual geographical area (NE Bulgaria), which would be way more informative.
Line 23: “their nocturnal activity” conveys the idea that the toad has a biphasic activity period, one diurnal and another nocturnal. I do not know this particular species, but I am well acquainted with their close relative, Pelobates cultripes, which is strictly nocturnal. If, as I suspect, P. balcanicus is chiefly nocturnal too, I would rephrase this into “their daily activity period, which is nocturnal”.
Line 24: A comma is needed after “toads”.
Lines 21-32: This part of the abstract is still messy and needs reorganization. It cannot go back and forth from methods to system description as it does. I suggest something like:

“We studied the thermal ecology of adult Balkan spadefoots (Pelobates balcanicus Karaman, 1928) in NE Bulgaria. These toads spend the daytime buried between 10 and 15 cm in sandy substrates, and emerge after sunset. On the substrate, their thermal energy exchange is defined by the absence of heat flow from the sun. Secondary heat sources like stored heat and infrared radiation from the soil play an important role for the thermal balance of the active spadefoot toads. At the beginning of their daily activity, we measured substrate temperature (at a depth of 11-12 cm) and toad surface body temperature, and also provided thermo-profiles of the animals and the substrate surface in their microhabitats. On the basis of our measurements and additional data, we discuss the eventual role of air humidity and the effects of surface and skin water evaporation on the water balance and activity of the investigated toads.”

Line 47: “ectothermic tetrapods”.
Lines 53-54: I see no need to specify this formula here. It is rather simple, it basically repeats the statement in the sentence, and thus it makes the reading more difficult.
Line 56: Delete “see” before “Mokhatla”).
Line 56: Actually, what is transferred is “heat”, not “temperature”.
Line 69: Delete “the” before “thermoregulation”.
Line 87: Change “to” with “with”.
Line 103: “We always followed”.
Line 106: Change “None” with “No”.
Lines 116-117: “on which a highly sensitive… is located”.
Line 130: Did you mean “unit”?
Line 154: “additionally measured”.
Line 178: Delete “measured to be”.
Line 195: “sections”.
Line 208: “did not use”.
Line 209: A comma is needed after “body temperature”.
Line 221: The comma after “means” is not correct.
Line 222: The comma after “precise” is not correct. What is more, this entire sentence should be rewritten, as it is awkward and its meaning is difficult to deduce.
Line 224: The comma after “demonstrates” should not be there.
Line 231: Change “big” for “large”. Except in some very specific fixed expressions, the word “big” should be avoided in scientific English, as it is somewhat colloquial.
Line 233: What do you mean “several second”? Did you want to say “few seconds”?
Line 237: “compared”.
Line 238: “surroundings”.
Line 242: Delete “the” before “environmental”.
Line 244: A comma is needed before “however”.
Line 245: “considered to be”.
Line 246: Delete “their”.
Lines 248-249: Did “cutaneous evaporation water loss” mean “cutaneous water evaporation”? Besides being grammatically awkward, the former wording is redundant”.
Line 253: Delete “the” before “water”.
Line 263: “the toad could also lose”.
Line 263: “(especially if the sky is clear)”.
Line 263: Delete the comma after “stressed”.
Line 265: “would only be possible”.
Line 265: Change “in case of” with “in the event of a”.
Line 270: Delete “The” before “surface”.
Line 271: Delete “the” before “spadefoot”.
Line 272: Delete “underground”.
Line 285: “sections”.
Line 294: Delete “the” before “relative”.
Line 298: change “and this” with “, which”.
Line 301: “those analyses,”.
Line 303: “. However,”.
Line 303: “conclusions”.

Experimental design

I have nothing new to add regarding the experimentalm design.

Validity of the findings

I have nothing to object to the validity of the findings.

---

## Round 0.3 · accepted · Accept

Thank you very much for the revision of the manuscript which is now also acceptable linguistically.